# *bglG* Regulates the Heterogeneity Driven by the Acid Tolerance Response in *Lacticaseibacillus paracasei* L9

**DOI:** 10.3390/foods12213971

**Published:** 2023-10-30

**Authors:** Zhichao Shen, Li Lin, Zhengyuan Zhai, Jingjing Liang, Long Chen, Yanling Hao, Liang Zhao

**Affiliations:** 1College of Food Science and Nutritional Engineering, China Agricultural University, Beijing 100083, China; shenzhichao1997@outlook.com (Z.S.); linli_derrick@163.com (L.L.); zhaizy@cau.edu.cn (Z.Z.); 15730088193@163.com (J.L.); shenzhenzhen1997@163.com (L.C.); 2Key Laboratory of Functional Dairy, Department of Nutrition and Health, China Agricultural University, Beijing 100193, China; haoyl@cau.edu.cn; 3Research Center for Probiotics, China Agricultural University, Sanhe 065200, China

**Keywords:** heterogeneity, acid tolerance response, lactic acid bacteria, *bglG*, subpopulation

## Abstract

The acid tolerance of lactic acid bacteria is crucial for their fermentation and probiotic functions. Acid adaption significantly enhances the acid tolerance of strains, and the phenotypic heterogeneity driven by the acid tolerance response (ATR) contributes to this process by providing a selective advantage in harsh environments. The mechanism of heterogeneity under the ATR is not yet clear, but individual gene expression differences are recognized as the cause. In this study, we observed four heterogeneous subpopulations (viable, injured, dead, and unstained) of *Lacticaseibacillus paracasei* L9 (L9) induced by acid adaption (pH 5.0, 40 min) using flow cytometry. The viable subpopulation represented a significantly superior acid tolerance to the injured subpopulation or total population. Different subpopulations were sorted and transcriptomic analysis was performed. Five genes were found to be upregulated in the viable subpopulation and downregulated in the injured subpopulation, and *bglG* (LPL9_RS14735) was identified as having a key role in this process. Using salicin (glucoside)-inducing gene expression and gene insertion mutagenesis, we verified that *bglG* regulated the heterogeneity of the acid stress response and that the relevant mechanisms might be related to activating *hsp20*. This study provides new evidence for the mechanism of the ATR and may contribute to the theoretical basis of improving the acid tolerance of *Lacticaseibacillus paracasei* L9.

## 1. Introduction

Microbial heterogeneity can be interpreted as phenotypic differences within a clonal, genetically identical population. It is a universal characteristic of living organisms [1]. The emergence of heterogeneous drug-resistant bacteria in clinical medicine is very common. For example, heteroresistant vancomycin-intermediate *Staphylococcus aureus* (hVISA, the hVISA strain is composed of subpopulations having different levels of vancomycin resistance), tigecycline heteroresistant *Acinetobacter baumannii* (AB), etc., have been widely reported [2,3]. Thus, clarifying the heterogeneity for the microbe stress tolerance and the elimination of pathogenic bacteria is important.

In many cases, heterogeneity can provide a selective advantage during environmental changes and stress. The formation mechanism of heterogeneity is not fully understood. Three models of population heterogeneity have been proposed thus far: biological (intrinsic) and environmental (extrinsic) models and a combined model [4,5]. Intrinsic heterogeneity can often arise due to cell cycle states, age distribution, or the stochasticity of gene expression and metabolic reactions [6]. For example, different cells may have different sizes and different numbers of key components, such as RNA polymerase, ribosomes, and small-molecule effectors [7]. Furthermore, the intrinsic stochasticity of gene expression leads to cell-to-cell fluctuations in mRNA molecules and key proteins [8,9,10]. This stochasticity is also known as the “noise” of cells [11]. The source of environment-driven heterogeneity is fluctuations in the environment. Following an environmental change, some individual transcriptional profiles are needed for survival in the new environment, which alters the spectrum of phenotypic members present within the population. Such behaviour may serve as a strategy and is essential for population growth and survival [12]. The fluctuations of gene expression programs are considered to cause phenotypic heterogeneity in clonal microbial populations.

Lactic acid bacteria (LAB) are commonly used in industrial production as food and beneficial supplements. However, acid stress is the main factor affecting the fermentation ability and activity of LAB. Heterogeneity is also found in LAB in response to environmental stress. Zhao et al. [13] studied the heterogeneity of *Lactobacillus brevis* exposed to beer compounds, finding an increased proportion of injured subpopulations, while the proportion of viable subpopulations decreased with increasing stress time. Papadimitriou et al. [14] obtained similar findings using *Streptococcus macedonicus* cells. Interestingly, the recovery of three subpopulations by single-cell sorting revealed that the viable subpopulation (cFDA+/PI−) exhibited the highest recovery (95 formed colonies out of 100 cells sorted), followed by the injured subpopulation (84/100), while no cell could recover from a dead subpopulation [14]. The heterogeneity found in *Lacticaseibacillus casei* [15], *Lactococcus lactis* [16], and *Lactobacillus* spp. [17] when exposed to the acid stress response may be explained as follows: (i) The distribution of subpopulations varies with Ph. (ii) There are significant differences in the culturability and acid tolerance between subpopulations. Among them, the viable subpopulation performed best, followed by the injured subpopulation, while the cell membrane and metabolic activity were somewhat damaged. (iii) Different subpopulations influence the acid tolerance of the whole population. However, the mechanism of heterogeneity induced by the acid tolerance response (ATR) is still not fully understood, and a theoretical basis for industrial production using heterogeneity to improve strain tolerance is lacking.

*Lacticaseibacillus paracasei* L9 is an aerotolerant Gram-positive bacterium belonging to the homofermentative lactic acid bacteria isolated from healthy people. *Lacticaseibacillus paracasei* L9 has been proven to modulate immunity and improve the intestinal barrier [18]. It has been industrialized and applied in products such as fermented milk and lactobacillus beverages. Previous studies have shown that the ATR can significantly improve the acid tolerance of the strain [19]. In this study, heterogeneous subpopulations of *Lacticaseibacillus paracasei* L9 were investigated after the ATR. Then, transcriptomic analysis was performed for the subpopulations, and the possible target genes regulating heterogeneity were analysed and verified. Finally, we determined the possible regulatory mechanism of heterogeneity regulation. This study will provide new evidence for the mechanism of the ATR and contribute to the theoretical basis of improving the acid tolerance of *Lacticaseibacillus paracasei* L9.

## 2. Materials and Methods

### 2.1. Bacterial Strains, Plasmids, and Growth Conditions

The bacterial strains and plasmids used in this study are listed in Appendix A. *Lacticaseibacillus paracasei* L9 was cultured in de Man–Rogosa–Sharpe (formula: peptone 10.0 g/L, beef extract 10.0 g/L, yeast extract powder 5.0 g/L, glucose 20 g/L, sodium acetate 5.0 g/L, ammonium citrate dibasic 2.0 g/L, K_2_HPO_4_·3H_2_O 2.0 g/L, MgSO_4_·7H_2_O 0.58 g/L, MnSO_4_·H_2_O 0.25 g/L, and tween 80 1.0 mL/L) medium with an initial pH between 6.4 and 6.6 at 37 °C. Colony-forming units (CFU) were counted after plating on MRS agar medium and incubation for 48 h at 37 °C. *Escherichia coli* (*E. coli*) was grown aerobically at 37 °C in Luria–Bertani (LB) medium with shaking. For selection, media were supplemented with antibiotics at the following concentrations: 5 μg/mL erythromycin (Solarbio, Bejing, China) in MRS for the *Lacticaseibacillus paracasei* L9 mutant and 100 μg/mL ampicillin (Solarbio, Beijing, China) in LB for *E. coli* DH5α. DL-lactic acid (Sigma, St. Louis, MO, USA) was added to adjust the pH (pH 6.5, pH 5.5, pH 5.0, pH 4.5, pH 3.5) of MRS. To investigate the influence of salicin on cell viability under acid stress, 25% glucose in MRS medium was replaced with an equal weight of salicin (Solarbio, Beijing, China). All results were obtained from three independent experiments.

### 2.2. Acid Adaption and Acid Challenge

Next, the impact of preincubation at different pH values on the survival ratio at the acid challenge (pH 3.5, 90 min) was tested. The exponential cells (OD_600nm_ = 0.6, 1.2 mL) were harvested by centrifugation (12,000× *g*, 5 min, 4 °C) and resuspended in isovolumetric MRS adjusted to pH 6.5, pH 5.5, pH 5.0, or pH 4.5 at 37 °C for 40 min. Subsequently, those acid-adapted cells were collected and incubated in MRS adjusted to pH 3.5 at 37 °C for 90 min (acid challenge). Plate count measured viable counts on MRS agar medium. Survival rates were calculated by dividing the number of CFU per milliliter after incubation at pH 3.5 by the number of cfu per milliliter after resuspension at pH 6.5 using the following formula:(1)$$\mathrm{Survival}\;\mathrm{rate}=\frac{cfu*{mL}^{-1}\;at\;pH\;3.5,\;90\;min}{cfu*{mL}^{-1}\;at\;pH\;6.5,\;90\;min}$$

### 2.3. Fluorescence Labelling and FCM Analysis

Cells were stained with two fluorescent dyes, SYTO13 (Invitrogen, S7575, Carlsbad, CA, USA) and propidium iodide (PI, Invitrogen, P3566, Carlsbad, CA, USA), as previously reported with slight modifications [20]. Samples from cultures were harvested by centrifugation (12,000× *g*, 5 min, 4 °C) and washed twice in phosphate-buffered saline (PBS, pH 7.4, sterile). Then, the cells in PBS were stained with SYTO13 at a final concentration of 1.5 μM for 15 min in the dark at 37 °C and then stained with 8.7 μM PI.

Gates in the flow cytometry (FCM) dot plots were established according to control samples: exponential cells (OD_600nm_ = 0.6) without any dyes; heat-killed cells treated at 121 °C for 15 min just with PI; exponentially growing cells with just SYTO13; and mixtures containing exponential and heat-killed cells (1:1) stained by PI and SYTO13.

Flow cytometry measurements were performed using a FACSCalibur (Becton Dickinson, Franklin Lakes, NJ, USA). Green fluorescence from SYTO13-stained cells was collected on the FL1 channel (530 ± 15 nm), whereas PI fluorescence was registered on the FL3 channel (>650 nm). For each analysis, 30,000 microspheres were acquired. The raw data were analysed with FlowJo v10 (Becton Dickinson, Franklin Lakes, NJ, USA). The results were obtained from three independent experiments.

### 2.4. Intracellular ATP Content

The intracellular ATP content was analysed according to the ATP Assay Kit (Beyotime, S0026, Beijing, China) following the manufacturer’s instructions. Luminance was measured by a microplate reader (Tecan, Infinite 200 PRO, Mnnedorf, Switzerland). Emitted light was linearly related to ATP concentration. Data were normalized to the control group and expressed as a percentage of control levels.

### 2.5. Transcriptomics

The exponential cells, cells after acid adaption, and different subpopulation cells (FCM-Viable, FCM-Injured, FCM-Total) were collected. Three independent biological replicates were performed in this study. Total RNA isolation was performed with TRIzol reagent (Invitrogen, Carlsbad, CA, USA) according to the manufacturer’s instructions. For RNA sample preparation, 3 μg RNA per sample was used as input material. Sequencing libraries were generated using the NEBNext Ultra Directional RNA Library Prep Kit for Illumina (NEB, Beverly, MA, USA) following the manufacturer’s recommendations, and index codes were added to attribute sequences. Library fragments were then purified using the AMPure XP system (Beckman Coulter, Beverly, MA, USA) to preferentially select cDNA fragments 150–200 bp in length. Size-selected and adaptor-ligated cDNA was then treated with 3 μL USER Enzyme (NEB, Beverly, MA, USA) at 37 °C for 15 min followed by 5 min at 95 °C. PCR was then performed using Phusion High-Fidelity DNA polymerase with universal PCR primers and Index (X) Primer. Finally, products were purified (AMPure XP system), and library quality was assessed on the Agilent Bioanalyzer 2100 system. The clustering of the index-coded samples was performed on a cBot Cluster Generation System using TruSeq PE Cluster Kit v3-cBot-HS (Illumina) according to the manufacturer’s instructions. After cluster generation, the library preparations were sequenced on an Illumina HiSeq platform, and paired-end reads were generated.

Raw data (raw reads) in fastq format were first processed through in-house Perl scripts. All sequenced reads were then aligned to the genome of *Lacticaseibacillus paracasei* L9. HTSeq v0.6.1 was used to count the read numbers mapped to each gene. Next, differential expression analysis of the two groups was performed using the DESeq R package (1.18.0). The resulting *p*-values were adjusted using Benjamini and Hochberg’s approach for controlling the false discovery rate. Genes with an adjusted *p*-value < 0.05 found by DESeq were considered differentially expressed.

All the raw transcriptomic data have been uploaded to NCBI (https://www.ncbi.nlm.nih.gov/sra, accessed on 3 February 2023) in BioProject (Accession No. PRJNA926023 and PRJNA921824).

### 2.6. Quantitative Real-Time PCR

RNA extraction was performed as described above. cDNA samples were obtained by reverse transcription of RNA samples with 5×All-in-One RT MasterMix (ABM, Vancouver, BC, Canada). Then, quantitative real-time PCR was performed using cDNA samples as templates. The primers of selected genes were designed based on the *Lacticaseibacillus paracasei* L9 genome using Primer-blast software (using Primer3 v4.1.0 and BLAST) in the online NCBI blast database and synthesized by Sangon Biotech, Shanghai, China. The primer pairs 16 s-F and 16 s-R, Q-*bglB*-F and Q-*bglB*-R, and Q-*bglG*-F and Q-*bglG*-R are listed in Appendix A. The reaction system of PCR was 20 μL:10 μL SYBR Green I real-time PCR Master Mix (Takara, Tokyo, Japanese), 1 μL cDNA template, 8 μL DNAse/RNAse-free water (Sigma, ST, Louis, MO, USA), 0.5 μL forwards primers and reverse primers. Real-time PCR was carried out with a Techne Quantica real-time PCR detection system (Techne, Minneapolis, MN, USA) under the following conditions: step 1: denaturation at 95 °C for 30 s; step 2: 40 cycles of 95 °C for 10 s and 60 °C for 30 s. Each reaction was performed in triplicate. The 16S rRNA gene was used as the internal reference gene [21].

### 2.7. DNA Manipulation Techniques

Chromosomal DNA from *Lacticaseibacillus paracasei* L9 was extracted using TIANamp Bacteria DNA Kit according to the manufacturer’s instructions (Tiangen, Beijing, China) [22]. Lysis of *Lacticaseibacillus paracasei* L9 was performed by adding 30 mg/mL lysozyme dissolved in TES buffer (50 mM Tris-HCl, 1 mM EDTA, 25% sucrose; pH 8.0) and incubating the suspension at 37 °C for 1 h. Miniprep plasmid isolation from *E. coli* was performed using the Plasmid Mini Kit I (OMEGA Biotek, Doraville, GA, USA). PCR was carried out using Q5 High-Fidelity DNA Polymerase (NEB, Beverly, MA, USA). The primers used in PCRs of the plasmid were M13/pUC sequencing primer, which is a universal primer of pUC19. Restriction endonuclease digestions were conducted according to the supplier’s instructions (Takara, Tokyo, Japan). DNA ligation was performed using the T4 DNA Ligation Kit (Thermo Fisher Scientific, Waltham, MA, USA). Plasmids were introduced into *E. coli* DH5α using standard heat shock transformation (Tiangen, Beijing, China). Plasmids were introduced into *Lacticaseibacillus paracasei* L9 using electroporation. DNA sequencing was performed with the BigDye Terminator cycle sequencing kit (Sangon, Beijing, China), and the results were further analysed with SnapGene (Dotmatics, England, UK).

### 2.8. Insertional Inactivation of bglG

To study the role of *bglG* in the acid tolerance response of *Lacticaseibacillus paracasei* L9, the *bglG* mutant of *Lacticaseibacillus paracasei* L9 was obtained by single crossover homologous recombination. An 890 bp internal region of *bglG* was chosen as a homologous sequence and amplified using the primer pair *bglG*-F and *bglG*-R. The resulting PCR product was restriction enzyme digested and ligated into the corresponding restriction sites of the suicide plasmid pUC19e. The recombinant plasmid, i.e., pUC*bglG*, was then introduced into *L. paracasei* L9 by electroporation, and the recombinant strain was cultivated on MRS solid medium containing erythromycin. As pUC*bglG* could not replicate in *Lacticaseibacillus paracasei* L9, the erythromycin selection pressure resulted in the integration of the plasmid into the *bglG* gene region of the genome. The resulting mutant was named L9*bglG*^−^. To confirm the integration of pUC*bglG* into the correct genome locus, PCR was performed with forwards primer T-*bglG*-F and reverse primer Em-R, which were designed according to the DNA sequence of the erythromycin resistance gene (GenBank Accession No. KM017875.1) and the downstream sequence of *bglG*, respectively. To maintain this type of mutation, antibiotics are needed in mutant strain activation. However, there is no antibiotic added when cultivating the working strain for proper control. The inserted gene can remain stable for 5 generations according to the PCR and sequencing results. The results are shown in Appendix A.

### 2.9. Statistical Analysis

The results are presented as the means ± standard errors of the means (SEM). Statistical differences were evaluated using one-way analysis of variance (ANOVA) or Student’s *t*-test at a significance threshold of 0.05. All of the data analyses were conducted using IBM SPSS Statistics v24 software (IBM, Armonk, NY, USA).

## 3. Results

### 3.1. Acid Tolerance Response of Lacticaseibacillus paracasei L9

We first tested the effect of different preincubation pH values on the survival of cells and found a pH range (5–5.5) that did not significantly affect the survival of *Lacticaseibacillus paracasei* L9 (Figure 1A). Then, the acid challenge (pH 3.5, 90 min) was performed after acid preincubation. The preincubation of *Lacticaseibacillus paracasei* L9 at pH 5.0 resulted in the highest survival rate in the acid challenge (18.1 ± 0.4%, Figure 1B). These results demonstrated that the acid tolerance response (ATR) could be triggered by transient exposure to a range of sublethal pH values. Thus, acid adaption (pH 5.0, 40 min) was used in subsequent experiments to characterize the ATR.

### 3.2. Heterogeneity Driven by the ATR of Lacticaseibacillus paracasei L9

The heterogeneity of *Lacticaseibacillus paracasei* L9 cells adapted at pH 5.0 for 40 min was analysed by flow cytometry staining with SYTO13 and propidium iodide (PI). SYTO13 and PI dyes are often used to assess viability by characterizing the integrity of cell membranes. When cells are simultaneously exposed to SYTO13 and PI, dead cells are usually PI+ and SYTO13−, while viable cells are usually PI− and SYTO13+. As shown in Figure 2A–C, unstained *Lacticaseibacillus paracasei* L9 live cells, heat-killed cells single-stained with PI, and live cells single-stained with SYTO 13 were used as control cells to divide different gates.

The ATR heterogeneity of *Lacticaseibacillus paracasei* L9 cells after acid adaption was observed, among which the subpopulation of viable cells was 87.1% ± 0.7%, and the subpopulation of injured cells was 11.5 ± 0.6% (Figure 2D and Appendix A). The total population (FCM-T), viable cells (FCM-V), and injured cells (FCM-I) were sorted and collected by flow cytometry sorting. Interestingly, the cells from different subpopulations showed diversity in acid tolerance when exposed to pH 3.5 for 20 min (Figure 2E). The viable subpopulation (FCM-V) showed the highest survival rate compared with the total population (FCM-T) and injured subpopulation (FCM-I), which revealed that the ATR in the viable subpopulation was significantly stronger than that in other subpopulations. Similarly, the intracellular ATP content of viable cells (FCM-V) was significantly higher than that in the total population (FCM-T), which might contribute to ATR (Figure 2F). These strategies are beneficial to *Lacticaseibacillus paracasei* L9 cells as they help them to survive better in a low-pH environment. The results suggest that *Lacticaseibacillus paracasei* L9 appears significantly heterogeneous after the ATR and that viable subpopulations have a greater impact on the improvement in acid tolerance.

### 3.3. Transcriptomic Changes in the ATR in Different Subpopulations

Regulation of gene expression was considered the main reason for microbial heterogeneity. To investigate genes involved in the heterogeneity of the ATR, differential gene expression among subpopulations was identified by RNA-Seq. The correlation of gene expression levels between three independent biological replicates was shown by the Pearson correlation coefficient and PCA (Appendix A). According to Figure 3, 17 genes related to the viable subpopulation and the injured subpopulation were obtained. Compared with samples without acid adaption, 5 genes were upregulated in the viable subpopulation but downregulated in the injured subpopulation, and 12 genes were downregulated in the viable subpopulation but upregulated in the injured subpopulation (Table 1). According to their putative functions, these genes could be involved in carbohydrate metabolism, nucleotide metabolism, translation processes, amino acid metabolism, stress response, etc.

It is worth noting that *bglG* (LPL9_RS14735), which encodes an anti-transcription terminator protein, was upregulated by 2.1-fold in the viable subpopulation and downregulated by 0.45-fold in the injured cell subpopulation (Table 1; Figure 4A). *bglG* is a structural gene of the *bgl* operon responsible for the translocation and hydrolysis of β-glucosides, such as salicin and arbutin [23,24,25]. The protein encoded by *bglG* has an antitermination effect [26,27,28]. Meanwhile, gene2284 (PTS beta-glucoside transporter subunit IIBCA) also has the same expression trend, indicating that β-glucoside’s metabolism might be involved in heterogeneous subpopulations (Table 1) [29]. In addition to *bglG*, the *bgl* operon contains *bglF* and *bglB* [30,31,32].

To further investigate the role of *bglG* in the ATR of *Lacticaseibacillus paracasei* L9, transcriptome data of *bglG* by RNA-Seq were verified by quantitative real-time polymerase chain reaction (qRT‒PCR). The gene expression of *bglG* was significantly upregulated after an acid adaption by 3.9-fold (Figure 4B).

### 3.4. Salicin Increases Acid Tolerance in Lacticaseibacillus paracasei L9

Salicin, a common kind of β-glucoside, was used to activate the *bgl* operon (*bglG*) and verify its impact on the subpopulation under acid stress. By replacing glucose with salicin in different proportions in the MRS medium, the survival rates of *Lacticaseibacillus paracasei* L9 to an acid challenge (pH 3.5, 90 min) were evaluated. Salicin was used to replace 25% glucose in the MRS medium due to it having the highest survival rate. The survival rate of the 25% salicin group was 17.1 ± 0.5%, which was significantly higher than that of the MRS group (13.7 ± 0.1%) (Figure 5A). The gene expression of *bglG* and *bglB* (the gene in the *bgl* operon) of *Lacticaseibacillus paracasei* L9 was increased when exposed to salicin, with *bglG* upregulated by 1.36-fold and *bglB* upregulated by 4.1-fold (Figure 5B).

By FCM, the proportion of viable subpopulations in the SAL group was 51.1 ± 0.6% after the acid challenge, which was significantly higher than that in the MRS group (37.7 ± 1.1%); the proportion of injured subpopulations in the SAL group was significantly reduced to 36.2% ± 0.4% compared with that in the MRS group (49.2 ± 1.2%) (Figure 5C,D). These results demonstrated that salicin induces the expression of *bglG* in *Lacticaseibacillus paracasei* L9, reducing the injured subpopulation and improving resistance during the acid challenge.

### 3.5. bglG Participates in the ATR Heterogeneity of Lacticaseibacillus paracasei L9

To further investigate the function of *bglG* in the heterogeneity of ATR in L9, a *bglG* mutant strain (L9bglG^-^) was constructed via insertion inactivation. The survival rates of L9bglG^−^ at pH 3.5 for 90 min sharply declined compared with those of the wild-type strain (WT). The survival rate of L9bglG^-^ was 3.5 ± 0.3%, which was 10.1% lower than that of the WT (Figure 6A). By FCM, a decrease in FCM-V and an increase in FCM-I in L9bglG^−^ after the acid challenge were observed. The viable subpopulation in L9bglG^-^ was reduced to 42.2 ± 0.4% compared with the WT (55.0 ± 1.5%). The FCM-I proportion in L9bglG^−^ was increased by 7.7% compared with that in WT (Figure 6B,C). These results revealed that the mutant strains were more seriously damaged during the acid challenge and that *bglG* may have important roles in the heterogeneity of the ATR. We supposed that *bglG* not only encoded the anti-transcriptional terminator and regulated the expression of *bgl* operon to enhance the utilization of β-glucoside but also regulated other downstream genes to participate in the heterogeneity of the ATR in *Lacticaseibacillus paracasei* L9.

### 3.6. Potential Genes Regulated by bglG in Acid Responses

To further investigate downstream genes regulated by *bglG*, gene expression among different subpopulations after the acid challenge (pH 3.5, 90 min) was identified by RNA-Seq. The correlation of gene expression levels between three independent biological replicates was analysed using the Pearson correlation coefficient (Appendix A).

Differential genes among different subpopulations were analysed according to the process shown in Appendix A. A total of 152 genes showed opposite regulation trends in L9bglG^-^ and L9-WT after the acid challenge (pH 3.5, 90 min), among which 52 genes were upregulated in L9-WT but not in L9bglG^-^, and 100 genes were downregulated in L9-WT but not in L9bglG^−^. In STEP 4 in Appendix A, a similar transcriptome analysis after the acid challenge in the SAL group was conducted, and two gene groups were identified.

Finally, we compared the results of STEP 4 with those of STEP 3, shown in Appendix A, to obtain the real downstream genes related to *bglG*. A total of 29 genes were upregulated after the acid challenge in SAL and WT but not in L9bglG^-^, and 5 had a more severe upregulation in SAL (Table 2). In addition, 79 genes were downregulated in SAL and WT but not in L9bglG^−^, and 48 were obviously downregulated in SAL (Appendix A). Among the five upregulated genes, LPL9_RS03660, described as an Hsp20/alpha-crystallin family protein, showed the highest fold upregulation Log_2_FC (SAL_ACID/SAL), followed by LPL9_RS05270 encoding a type II secretion system F family protein; LPL9_RS08570, related to a hypothetical protein, LPL9_RS10845, described as a MarR family transcriptional regulator; and *rpsB,* supposed to express the 30S ribosomal protein S2.

## 4. Discussion

In this study, we observed a heterogeneous subpopulation of *Lacticaseibacillus paracasei* L9 after acid adaption (pH 5.0, 40 min) by FCM. Different subpopulations showed varying survival rates after exposing cells to acid stress, while viable cells (FCM-V) represented a higher survival rate than injured cells (FCM-I) or the total population. Transcriptome analysis revealed that *bglG* in the *bgl* operon (β-glucosidase) was upregulated in FCM-V cells but downregulated in FCM-I cells. Using β-glucoside (salicin) induction and gene mutants, we confirmed that *bglG* could regulate the heterogeneity of ATR in *Lacticaseibacillus paracasei* L9. Five genes, including *hsp20*, might be involved in the regulation of ATR heterogeneity by *bglG.* To the best of our knowledge, this is the first study reporting on the mechanism of heterogeneity under the ATR in lactic acid bacteria.

Heterogeneity is an effective strategy for microorganisms to fight environmental changes. Diversity within a population can greatly benefit the group, especially when the population experiences sudden changes. In the present study, we found that the acid resistance of *Lacticaseibacillus paracasei* L9 increased after acid adaption (pH 5.0, 40 min), and there was no significant difference in bacterial number under plate culture but different subpopulations appeared. Similarly, Zhao et al. [13] found that the percentages of injured and dead cells increased, while viable cells decreased in *Lactobacillus brevis* with increasing stress time when exposed to beer compounds. Similar results were found in *Lactococcus lactis* during pH-controlled fermentation [16] and *Lactobacillus* spp. when pH changed [17]. At the same time, we detected that the survival rate of the total population (FCM-T) was between the FCM-V and FCM-I subpopulations, which confirmed that the acid resistance of the total population was a comprehensive reflection of different subpopulations. There is limited research on the subpopulations and tolerances in LAB. Subpopulations differed in metabolic activity: the injured and dead cells were revealed as nonproducing lactic acid cells in *Lacticaseibacillus casei* [15]. Remarkably, exposure to pH 5.0 resulted in the existence of “injured” cells, which could partly recover by plate culturing but with lower metabolic activity [14,17]. The results indicate the overestimation of the acid tolerance of strains in ordinary analysis. Similar results were also observed in *Lactobacillus brevis* under beer stress, and “viable” cells showed stronger tolerance [13,14]. Research on the mechanism of heterogeneity formation in lactic acid bacteria is still limited, especially the process of heterogeneity formation under environmental pressure. In addition to LAB, heterogeneous subpopulations of pathogenic bacteria were also found to be resistant to antibiotics in *Mycobacterium tuberculosis* cells [33,34]. This study showed that the distribution of subpopulations determined the acid tolerance of *Lacticaseibacillus paracasei* L9, suggesting that the acid tolerance of the strain could be improved by increasing the viable subpopulation and reducing injured subpopulation.

Heterogeneity is an inherent characteristic of microorganisms. However, the mechanism of its formation in LAB is not fully understood. The differences in gene expression in heterogeneous subpopulations driven by stochastic fluctuations or environmental changes were considered an important mechanism. Qi et al. [35] observed obvious gene expression heterogeneity between single cells in biofilm and planktonic culture in *Desulfovibrio vulgaris* by single-cell qRT‒PCR analysis. In the present study, different subpopulations of *Lacticaseibacillus paracasei* L9 were sorted by FCM, and RNA-Seq was used to analyse the differences in gene expression among subpopulations. A total of 17 genes were significantly changed in acid-derived subpopulations. LPL9_RS10760 (PTS beta-glucoside transporter subunit IIBCA), LPL9_RS14735 (BglG family transcriptional antiterminator), and LPL9_RS00305 (ABC transporter permease) were all related to glucoside metabolism and upregulated in the viable subpopulation. These results all point to the *bgl* operon. Therefore, this study focused on the role of the *bgl* operon in subpopulation regulation. Some tolerance-related proteins in the viable subpopulations, such as LPL9_RS14325 and LPL9_RS03660, had decreased expression, indicating that the gene expression in the subpopulations was different from that in the population. In addition, gene 59 (transcriptive regulator), with the lowest expression, may have been suppressed in the viable subpopulation, but its function needs to be studied in the future. Interestingly, after acid adaption (pH 5.0, 40 min), the *bglG* expression of the β-glucosidase operon was significantly upregulated in the viable cells (FCM-V) but also downregulated in the injured cells (FCM-I). Using β-glucoside (salicin)-induced *bgl* operon (*bglG*) gene expression, the proportion of FCM-V increased significantly, and FCM-I decreased with improved acid tolerance of the strain. Using the *bglG^-^* mutant, the proportion of FCM-V decreased significantly, with the acid tolerance of L9bglG^-^ declining. These results confirmed that *bglG* has an important role in the ATR heterogeneity of *Lacticaseibacillus paracasei* L9. We found that target genes regulate the ATR heterogeneity of LAB, providing new evidence and methods for acid tolerance regulation in LAB.

The *bglG* in the *bgl* operon has a key role in regulating the ATR heterogeneity of *Lacticaseibacillus paracasei* L9. There is little research on the environmental tolerance of *bgl* in microorganisms, and currently, only its structure and function have been demonstrated in *E. coli*. There are few reports that *bgl* directly affects the tolerance of bacteria. The *bgl* operon is recognized in the process of utilizing the β-glucosides, such as salicin. In *E. coli*, the *bgl* operon comprises three structural genes, *bglG*, *bglF,* and *bglB,* and a regulatory region *bglR*. *bglG* encodes an antiterminator. *bglF* encodes a PTS permease and a negative regulator of the *bgl* operon, while *bglB* encodes a phospho-β-glucosidase [27]. During evolution, the *bgl* operon remained silent in *E. coli*, but there was no accumulation of inactivation mutations [28]. One explanation for the organism retaining this recessive gene was that although the gene was silent under laboratory conditions, it could be expressed under specific conditions, providing selective advantages for the organism [27]. Therefore, it is speculated that the *bgl* operon might be involved in the regulation of other cellular functions [23,31,36]. However, no direct evidence suggested that *bgl* was related to the heterogeneity or ATR of microorganisms. Madan et al. found that the *bgl* operon of *E. coli* confers a growth advantage in the stationary phase (GASP), which indicated that wild-type *E. coli* retaining the *bgl* operon might have an evolutionary advantage [37]. Harwani et al. discovered that the *bgl* operon had an indirect regulatory effect on the downstream *oppA*-encoding oligonucleotide peptide transporter subunits [38]. OppA assists in transporting small peptide segments, enabling potential nutrient absorption and giving *E. coli bgl^+^* strain growth advantages in the stationary phase. In addition to its role in β-glucoside catabolism, the *bgl* operon also regulates other downstream target genes, forming a selective advantage for organisms. Houman et al. proposed that *bglG* could combine with the RAT sequence and regulate the transcription of downstream target genes with similar RAT sequences [39]. Gulati et al. found that *bglG* could affect gene expression by stabilizing the mRNA [40]. In this study, using mutants and salicin induction, we first verified that *bglG* regulated the ATR heterogeneity of *Lacticaseibacillus paracasei* L9, which may be related to *bgl*-target downstream gene expression.

*bglG* could improve the tolerance of individual cells by regulating their downstream genes. The PTS system may increase different sugar utilization functions and regulate tolerance-related genes to improve environmental tolerance. LPL9_RS03660, encoding an Hsp20/alpha-crystallin family protein, may be involved. The heat shock protein Hsp20 is a family of heat shock proteins synthesized in response to heat or other environmental stresses that act as protein chaperones that can protect other proteins against denaturation and aggregation [41]. We did observe a decrease in gene expression of Hsp20 protein (LPL9_RS03660) in the FCM-V subgroup, while an increase in gene expression of this protein was observed in the FCM-I subgroup. However, we also observed a significant upregulation of Hsp20 protein-encoding gene expression in the total population (FCM-T) of the L9 strain after acid adaption. In subsequent studies, we found that the gene expression of Hsp20 protein in the total population was significantly upregulated after treatment with sublethal acid (acid challenge). The expression level was further slightly upregulated after salicin-induced *bglG*, while this effect was weakened after *bglG* deletion. This indicates that Hsp20 may be regulated by BglG. The gene expression levels of Hsp20 in different subpopulations cannot be simply statistically analysed. Heterogeneity within bacterial populations showed different trends from the averaging effect of bulk measurements and could be different from the binary logic imposed by traditional culture-based techniques [14]. Hsp20 has been partially reported in the environmental stress response of lactic acid bacteria, such as *Bifidobacterium breve* [42], *Lactobacillus paracasei* [43], and *Lactobacillus acetotolerans* [44]. It might be associated with early translation and trafficking of stored mRNAs during the exponential phase [44]. However, further research is needed on the role of Hsp20 in the heterogeneity of the ATR. In addition, the MarR (multiple antibiotic resistance repressor) family transcriptional regulator encoded by the gene LPL9_RS10845 regulates antibiotic resistance within a group of pathogens and is related to controlling the oxidative stress response and biofilm formation [41,45]. The results suggested that in addition to the use of salicin, *bglG* also regulated other downstream target genes encoding Hsp20 and MarR family transcriptional regulators, which are associated with the environmental stress response of microbes. The relationship between the *bgl* operon and subpopulations and how the *bgl* operon could regulate its downstream genes need to be further studied at the single-cell level. Meanwhile, gene annotation in *Lacticaseibacillus paracasei* L9 indicated that *bglG* is also the homeotic gene of mannitol-specific transcription activator (*mtlR*). The insertion inactivation of *bglG* possibly results in polarity and affects the transcription of its downstream genes (*mtlF*, encoding mannitol-specific IIBC components, LPL9_RS14730; *mtlD*, encoding mannitol-1-phosphate 5-dehydrogenase, LPL9_RS14725), which might be related to the metabolism of mannitol [21]. Future research needs to further elucidate the relationship between *bglG* and its downstream genes, as well as its role in the heterogeneity of ATR.

## 5. Conclusions

This study found that acid adaption-induced ATR significantly improved the acid tolerance of *Lacticaseibacillus paracasei* L9 at the population level. More importantly, we observed that the ATR induced heterogeneous subpopulations of *Lacticaseibacillus paracasei* L9, with the live subpopulation having the highest acid tolerance, and the acid tolerance of the total population was a manifestation of the subpopulations. Furthermore, we found for the first time at the microbial subpopulation level that *bglG* plays a crucial role in the ATR heterogeneity of *Lacticaseibacillus paracasei* L9, and its downstream regulation of *hsp20* may be an important regulatory mechanism. This research confirmed that gene expression fluctuations can cause heterogeneity in environmental stress tolerance in lactic acid bacteria, and by locating and regulating genes related to heterogeneity, a new perspective was proposed to improve the stress tolerance of strains.

## Figures and Tables

**Figure 1 foods-12-03971-f001:**
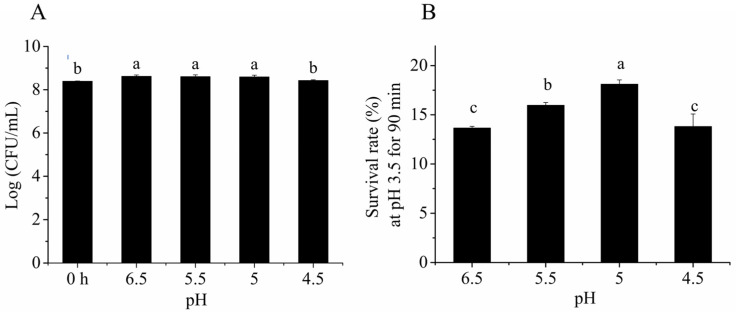
Survival of *Lacticaseibacillus paracasei* L9 under acid challenge following acid adaption with different pH. (**A**) Acid treatment at pH 5.0 for 40 min did not significantly affect its survival. The 0 h bar represents the viable count of *Lacticaseibacillus paracasei* L9 before acid treatment. (**B**) Acid adaption at pH 5.0 significantly increased the survival rate of *Lacticaseibacillus paracasei* L9 under a lethal acid challenge (pH 3.5 for 90 min). a, b, c indicate significant differences (*p* < 0.05). Data are presented as the mean ± SD from three biological replicates.

**Figure 2 foods-12-03971-f002:**
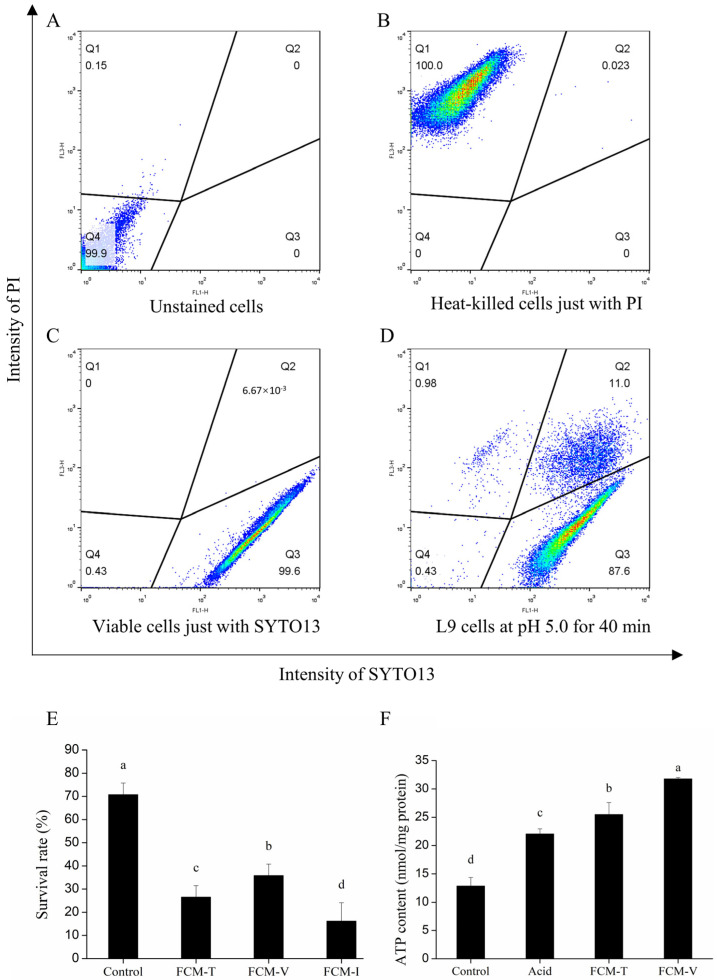
Heterogeneity and subpopulation physiology induced by acid tolerance response of *Lacticaseibacillus paracasei* L9. (**A**–**D**) Distribution of *Lacticaseibacillus paracasei* L9 cells within a flow cytometric dot plot, gate Q4: SYTO13−/PI−, double-negative cells including fragments and unstained intact cells (**A**); gate Q1: SYTO13−/PI+, dead cells (**B**); gate Q2: SYTO13+/PI+, injured cells; gate Q3: SYTO13+/PI−, intact viable cells (**C**). Based on the region of interest with SYTO13 and PI fluorescence, cells were sorted into three groups by flow cytometry sorting: viable subpopulation (FCM-V), injured subpopulation (FCM-I), and total bacteria population (FCM-T). (**E**,**F**) Properties of acid resistance of different subpopulations were evaluated. The survival rate of the FCM-V subpopulation was significantly higher than that of the others at pH 3.5 for 20 min (**E**). Furthermore, different subpopulations had significant differences in intracellular ATP content (**F**). a, b, c, d indicate significant differences (*p* < 0.05).

**Figure 3 foods-12-03971-f003:**
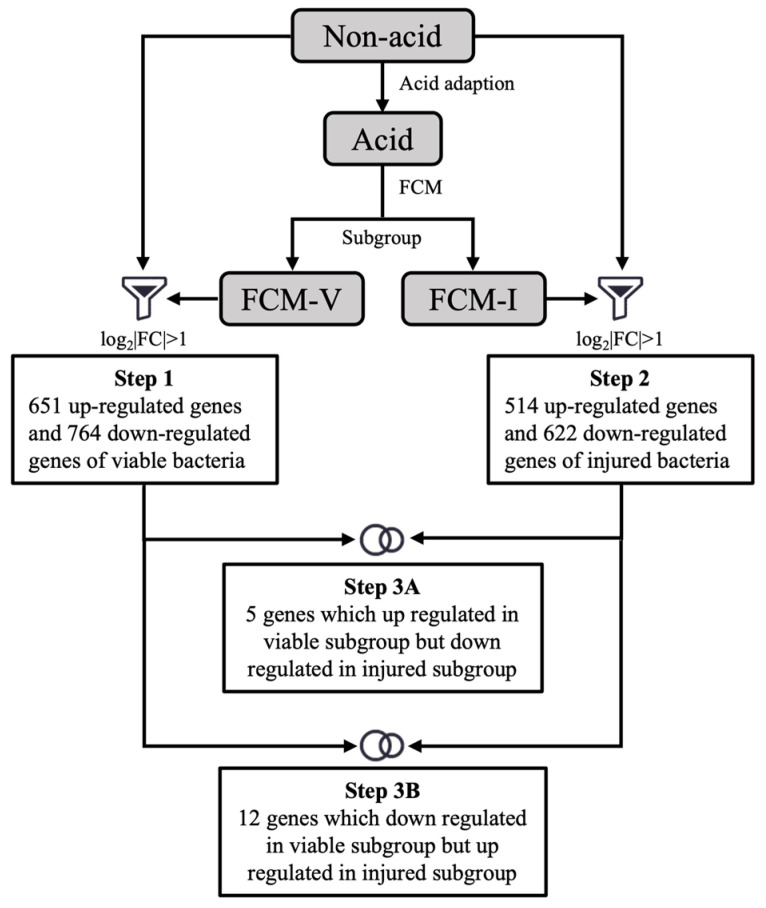
The analysis of heterogeneity of ATR genes of *Lacticaseibacillus paracasei* L9 by RNA-Sequencing. The transcriptome data were screened and analysed, and 17 genes that showed different regulatory trends between the FCM-V subpopulation and FCM-I subpopulation compared with the non-acid group were selected.

**Figure 4 foods-12-03971-f004:**
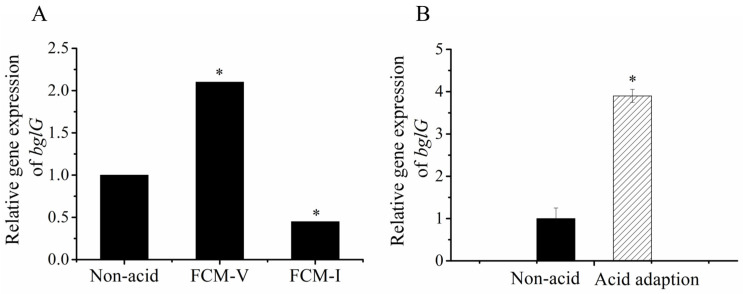
Relative gene expression of *bglG* in different subpopulations. (**A**) By analyzing the transcriptome data, the relative expression of *bglG* showed an opposite trend in FCM-V and FCM-I. (**B**) Subsequently, the relative expression of *bglG* after acid adaption (pH 5.0 for 40 min) was verified by RT‒PCR. * indicates significant differences (*p* < 0.05).

**Figure 5 foods-12-03971-f005:**
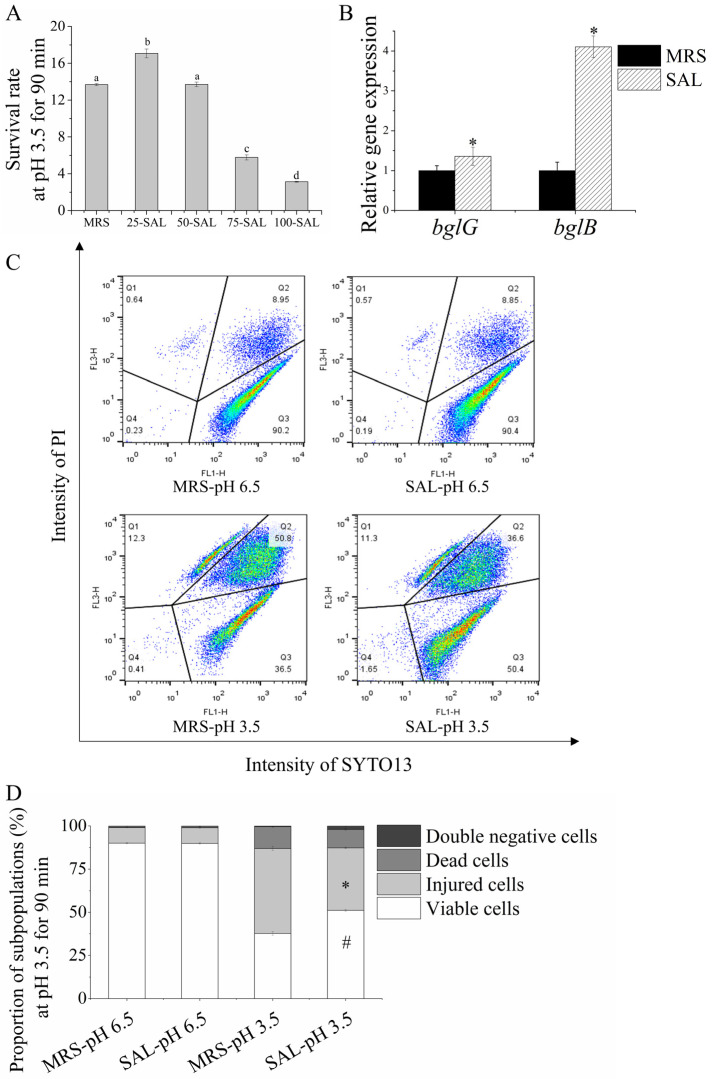
Effects of salicin on acid tolerance characteristics of *Lacticaseibacillus paracasei* L9. (**A**) The survival rate at pH 3.5 for 90 min using salicin to replace glucose in different proportions. a, b, c, d indicate significant differences (*p* < 0.05). (**B)** The expression of *bglG* and *bglB* was induced by salicin. (**C**,**D**) The proportion of viable subpopulations of *Lacticaseibacillus paracasei* L9 at pH 3.5 for 90 min was effectively improved. In flow cytometry dot plot (**C**), Q1: dead cells (SYTO13−/PI+), Q2: injured cells (SYTO13+/PI+), Q3: viable cells (SYTO13+/PI−), Q4: double-negative cells including fragments and unstained intact cells (SYTO13−/PI−). * and # indicate significant differences between the SAL-pH 3.5 group and MRS-pH 3.5 group in the proportion of injured and viable subpopulations, respectively (*p* < 0.05).

**Figure 6 foods-12-03971-f006:**
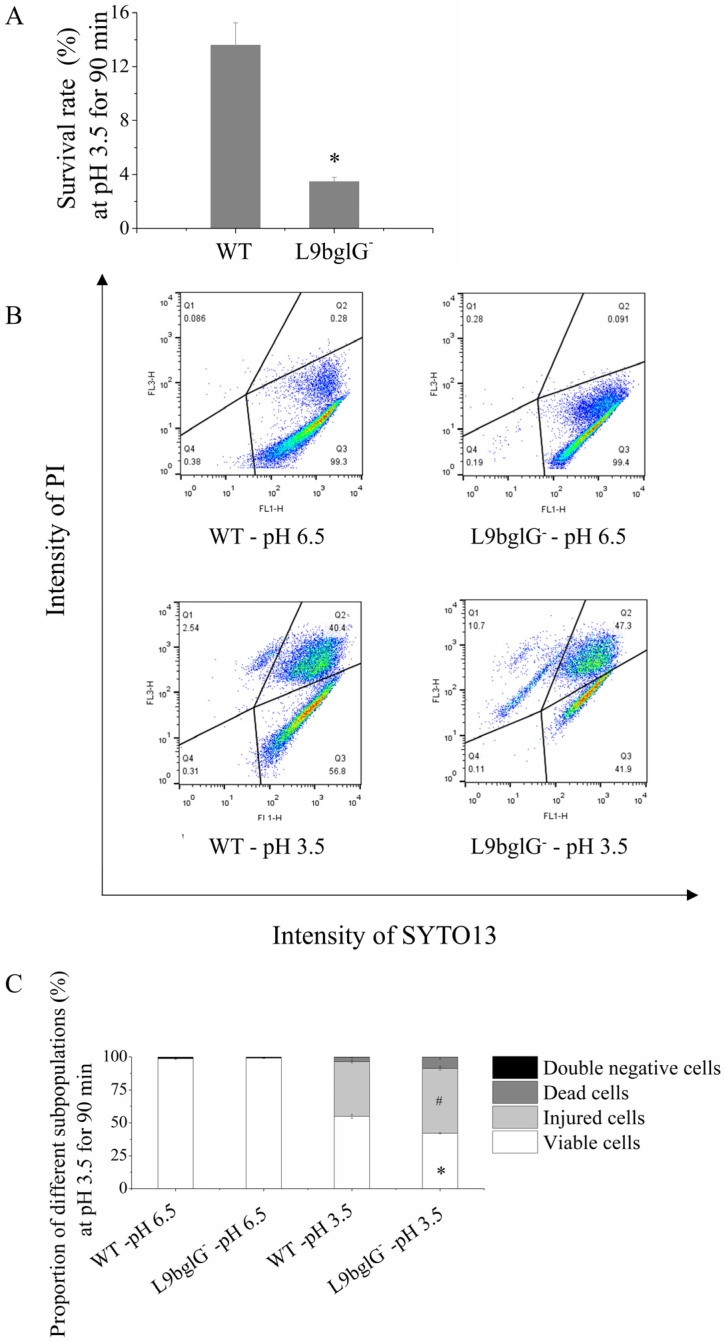
Effects of *bglG* on acid tolerance characteristics of *Lacticaseibacillus paracasei* L9. (**A**–**C**) The survival rate (**A**) and proportion of different subpopulations (**B**,**C**) of L9bglG^-^ after acid challenge (pH 3.5 for 90 min). In the flow cytometry dot plot (**B**), Q1: dead cells (SYTO13−/PI+), Q2: injured cells (SYTO13+/PI+), Q3: viable cells (SYTO13+/PI−), Q4: double-negative cells including fragments and unstained intact cells (SYTO13−/PI−). * and # indicate significant differences among the mutant and wild-type strains, respectively, in the proportion of viable and injured subpopulations (*p* < 0.05).

**Table 1 foods-12-03971-t001:** A total of 17 genes showed different regulatory trends between the FCM-V subpopulation and FCM-I subpopulation compared with samples without acid adaption.

No.	Gene ID	Gene Name	log_2_FC (FCM-V/Non-Acid)	log_2_FC (FCM-I/Non-Acid)	Annotated Function
1	gene2198	LPL9_RS10370	2.0377 ↑	−1.0162 ↓	Amino acid ABC transporter permease
2	gene2284	LPL9_RS10760	1.6958 ↑	−1.3129 ↓	PTS beta-glucoside transporter subunit IIBCA
3	gene807	LPL9_RS03780	1.275 ↑	−1.0363 ↓	Iron–sulfur cluster biosynthesis family protein
4	gene66	LPL9_RS00305	1.2064 ↑	−1.0832 ↓	ABC transporter permease
5	gene3139	LPL9_RS14735	1.0483 ↑	−1.1387 ↓	BglG family transcriptional antiterminator
6	gene59	LPL9_RS00270	−3.3762 ↓	1.0407 ↑	Transcriptional regulator
7	gene1874	LPL9_RS08830	−2.8654 ↓	1.156 ↑	Hypothetical protein
8	gene2556	LPL9_RS12000	−2.7301 ↓	1.2204 ↑	Glutamate 5-kinase
9	gene780	LPL9_RS03660	−1.9645 ↓	1.2441 ↑	Hsp20/alpha-crystallin family protein
10	gene3020	LPL9_RS14160	−1.7589 ↓	1.0555 ↑	NUDIX domain-containing protein
11	gene3055	LPL9_RS14325	−1.5069 ↓	1.2373 ↑	Hsp20/alpha-crystallin family protein
12	gene1995	LPL9_RS09420	−1.4664 ↓	1.1811 ↑	N-acetyltransferase
13	gene1535	LPL9_RS07225	−1.4 ↓	1.0644 ↑	Hypothetical protein
14	gene1231	LPL9_RS05775	−1.2073 ↓	2.3676 ↑	Flavodoxin
15	gene1741	LPL9_RS08205	−1.1713 ↓	1.0484 ↑	Zinc-containing alcohol dehydrogenase/quinone oxidoreductase[NADPH]
16	gene28	LPL9_RS00140	−1.1 ↓	1.9574 ↑	Hypothetical protein
17	gene1239	LPL9_RS05810	−1.0844 ↓	1.3583 ↑	DEAD/DEAH box helicase

↑ indicates upregulate and ↓ indicates downregulate.

**Table 2 foods-12-03971-t002:** Five selected genes upregulated after acid challenge in SAL and WT but not in L9bglG^-^, more obviously upregulated in SAL.

Gene ID	Gene Description	Log_2_FC (SAL_ACID/SAL)		Log_2_FC (WT_ACID/WT)
LPL9_RS03660	Hsp20/alpha-crystallin family protein	2.11	>	1.99
LPL9_RS05270	Type II secretion system F family protein	1.61	>	1.29
LPL9_RS08570	Hypothetical protein	1.21	>	1.13
LPL9_RS10845	MarR family transcriptional regulator	1.12	>	1.04
LPL9_RS08520	30S ribosomal protein S2	1.1	>	1.08

## Data Availability

All the transcriptomics raw data have been uploaded to NCBI (https://www.ncbi.nlm.nih.gov/sra, accessed on 3 February 2023) in BioProject (Accession No. PRJNA926023 and PRJNA921824).

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
