# Peer review of "bglG Regulates the Heterogeneity Driven by the Acid Tolerance Response in Lacticaseibacillus paracasei L9"

_foods, 2023, doi:10.3390/foods12213971_

Round 1

Reviewer 1 Report

Comments and Suggestions for Authors

General comments:

The paper's subject has novelty and interest. There are big knowledge advancements. However, the authors should make some changes in the manuscript to be accepted. I have a few comments about the Abstract, Introduction, Results and Discussion (listed below).

Specific points:

-Please carefully review the English across all manuscript. For example, please replace “Five genes were founded” in the abstract section for “Five genes were found”.

- More information about the possible applications and importance of LAB and L9 should be introduced in the “Introduction” section.

-Please check lines 86-87; 165; 205; 239; 353; 356; 366:” Error! Reference source not found.”

-In line 112 and 357 please gives space in “pH7.4” andpH3.5”. Please revise similar mistakes.

-Figure 1 and 4: Improve their resolution; the captions should be smaller ( Figure 1 and 4) and pH as the axis title instead of pH0, pH6.5, etc. and use space after “log” and “survival rate” (Figure 1).

-Please check the unit format/space before the unit across the manuscript (for example, line 225- 0h).

- Figures 2, 5 and 6 are barely noticeable, please increase the size and resolution. The axis titles are not visible, especially in Figure 2 A, B, C and D as well as Figure 5 c and Figure 6b.

- The results presented in Table 1 should be more discussed in the text.

-In section 4 additional studies with other LAB species should be compared for a deeper and additional discussion.

- The citations throughout the text and references section should be carefully reviewed. This section should be checked as indicated in the guide for authors of the Journal.

Comments on the Quality of English Language

Please carefully review the English across all manuscript. For example, please replace “Five genes were founded” in the abstract section for “Five genes were found”.

Reviewer 2 Report

Comments and Suggestions for Authors

Review:

Authors present a work where they evaluate the role of the blg operon in Lacticaseibacillus paracasei by using cell sorting and mutants.

The work is interesting and provide new data into the field, specially by the use of cell sorting and mutant strategies to study the acid tolerance and adaptation in lactic acid bacteria. However, M&M section should be clearer and more detailed to really understand the work done. Some crucial information is missing. Also, repetitive format and typing errors are found within the text. Besides, Figures should be specially redone to increase the impact of the research. Finally, discussion section is very general about the possible alternative role of the studied operon in the species E. coli, but not really focused in this bacterium or even the provided results.

Some specific comments:

L76: specify the species of the strain. Here and now on. I can imagine that for the authors is obvious the species they work with. However, it is essential to repeat this information within the manuscript to help the reader follow the text.

L87 Please check the error. Here and now on.

L88 Specify the species

L94 D and L isomers?

L95 which pH values?

L96 so the resulting MRS is not a commercial powder. Then, list the composition.

L97 Please explain the meaning of "at least"

L99 what is challenge pH?

L101 volume of the culture? Please provide more information about the culture conditions.

L114 the acronym is then introduced. Check these kind mistakes.

L123 replicates in the technique?

L129. Transcriptomics of which samples?

L130 Early, mid... exponential phases. Please, explain it. The samples are taken in different stages. This is essential to interpret the data.

L165 information is missing.

L171 no replicates of each reaction. Just one reference gene?

L204 how was this calculated?

L213 the paragraph is confusing. It should be rewritten.

Figure 1. Carefully describe the figure. Figure 1A is not a survival rate.

Figure 2. Figure caption is repetitive. It is very similar to the main text. Also, do not change the order of the data in the histograms. You should maintain the same order to easily interpret the data.

Table 1. the classification of these genes in orthologous groups should also be interesting.

Table 1. How can you explain that the hsp proteins are downregulated in viable cells?

Figure 5 and 6. I think the figures are redundant. Figure C is the summary of B in another representation. Please check this and reduce redundancy here and in the other figures.

L359 these genes should be addressed, at least as supplementary material.

What is STEP 4?

Table 2. The nomenclature is not clear. I suggest trying to unify and making it clearer.

L381 hsp20 is downregulated in these viable cells. I do not see the link between a higher survival and a lower hsp20 expression.

Discussion is very general with no direct link with the results provided. Also, a lot of dissertation about the role of the bgl operon in E. coli.
